# Short-Time Impact of Soil Amendments with *Medicago* Plant Materials on Soil Nematofauna

**DOI:** 10.3390/plants10010145

**Published:** 2021-01-12

**Authors:** Marek Renčo, Nikoletta Ntalli, Trifone D’Addabbo

**Affiliations:** 1Institute of Parasitology, Slovak Academy of Sciences, Hlinkova 3, 04001 Košice, Slovakia; 2Department of Pesticides’ Control and Phytopharmacy, Benaki Phytopathological Institute, 8 S. Delta Str., 14561 Athens, Greece; nntali@agro.auth.gr; 3Institute for Sustainable Plant Protection—CNR, 70126 Bari, Italy; trifone.daddabbo@ipsp.cnr.it

**Keywords:** *Medicago* species, plant biomasses, soil amendments, nematode community soil food web descriptors

## Abstract

Soil amendments with plant materials from *Medicago* species are widely acknowledged for a suppressive effect on plant-parasitic nematodes but their impact on beneficial components of soil nematofauna is still unknown. A study on potted tomato was carried out to investigate the short-time effects on the overall nematofauna of dry biomasses from six different *Medicago* species, i.e., *M. sativa*, *M. heyniana*, *M. hybrida*, *M. lupulina*, *M. murex* and *M. truncatula*, incorporated to natural soil at 10, 20, or 40 g kg^−1^ soil rates. All amendments resulted in a significant decrease of the total nematofauna biomass, whereas total abundance was significantly reduced only by *M. heyniana*, *M. hybrida*, and *M. lupulina* biomasses. Almost all the *Medicago* amendments significantly reduced the relative abundance of plant-parasites and root fungal feeders. All amendments significantly increased the abundance of bacterivores, whereas fungivores significantly increased only in soil amended with *M. heyniana*, *M. lupulina* and *M. sativa* plant materials. *Mesorhabditis* and *Rhabditis* were the most abundant genera of bacterivores, whereas *Aphelenchoides* and *Aphelenchus* prevailed among the fungivores. Predators were poorly influenced by all the tested *Medicago* biomasses, whereas the abundance of omnivores was negatively affected by *M. heyniana* and *M. lupulina*. Values of the Maturity Index and Sum Maturity Index were reduced by treatments with *M. heyniana*, *M. hybrida*, *M. lupulina* and *M. sativa* plant materials, whereas most of the tested amendments decreased values of the Channel Index while increasing those of the Enrichment Index. Enrichment and bacterivore footprints raised following soil addition with *Medicago* biomasses, whereas composite and fungivore footprints were significantly reduced. According to their overall positive effects on soil nematofauna, amendments with *Medicago* plant materials or their formulated derivatives could represent an additional tool for a sustainable management of plant-parasitic nematodes.

## 1. Introduction

The economic impact of plant-parasitic nematodes on agricultural crops imposes the use of control measures aiming to reduce their soil population densities under damage thresholds [1]. The EU (European Union) restrictions of the use of plant protection products (Reg. EC 889/2008; 128/2009; 1185/2009) [2] has given a strong impulse to research on safer control alternatives to synthetic nematicides, including the use of soil organic amendments based on agroindustrial wastes, crop residual biomasses, or green manures [3,4]. The suppressiveness of these materials to phytonematodes could involve several concurrent or alternative mechanisms, such as the stimulation of soil nematode- antagonistic microflora or the release of nematicidal compounds [5,6], but is still not clearly elucidated.

Soil incorporation of fresh or dry biomasses from several plants has been frequently proven to have a suppressive effect on various plant-parasitic nematodes, such as root-knot species (*Meloidogyne* spp.) cyst-forming nematodes (*Globodera* spp., *Heterodera* spp.) or ectoparasite species (*Pratylenchus* spp., *Xiphinema index*) [7,8,9,10,11,12,13]. In particular, soil treatments with dry biomasses from *Medicago* species repeatedly resulted in a strong reduction of root-knot and cyst nematode infestations both in pot and field experiments [14,15,16,17], mainly due to the high content of preformed nematotoxic saponins [18] and to the release of toxic ammoniacal compounds by tissue degradation into the soil [19,20]. 

In addition to plant parasitism, components of soil nematofauna can show several more feeding behaviors, according to which they are usually classified into five basic trophic groups, i.e., bacterivores, fungivores, omnivores, predators, and plant parasites [21]. All these trophic groups give a relevant contribution to soil food web functions, driving important soil ecosystem processes (microbial spread, organic matter decomposition, nitrogen mineralization, and more) [22,23,24,25,26,27]. 

Abundance, ubiquitous distribution, and a close relationship with other soil food web components make nematodes able to promptly respond to the environmental changes and, therefore, represent affordable bioindicators of any perturbation occurring in soil [23,24,27,28,29,30]. According to nematode trophic groups’ response to variations of soil conditions, a number of ecological indices have been developed, such as the maturity (MI), enrichment (EI), structure (SI), and channel (CI) index as well as the metabolic footprints [24,31,32]. These indices allow the monitoring of soil health conditions following any ecosystem disturbance, including agronomical practices such as organic amendments´ soil incorporation [33,34,35]. 

Components of soil nematofauna were found to be modified after soil amendments with plant biomasses. Such effects on the abundance and presence of particular trophic groups however depend on the amount and composition of the amendment raw materials. Populations of bacterivore and fungivore trophic groups were found to increase after soil amendments with different raw or composted residues of cotton (*Gossypium hirsutum* L.) and rye (*Secale cereale* L.) mixed to vetch (*Vicia sativa* L.) [36], or maize (*Zea mays* L.), Texas panicum (*Panicum texanum* Buckley R. Webster) and velvet bean (*Mucuna pruriens* L. DC.) [37]. A bacterial decomposition pathway was also observed following soil incorporation with green biomasses of Brassicaceae plants such as yellow mustard (*Sinapis alba* L.) and radish (*Raphanus sativus* L.) [38], whereas a greater fungal-based food web occurred in soil amended with rapeseed (*Brassica napus* L.) and rye, constantly with no changes in omnivore and predator abundance [39]. There is no previous study on the impact of soil amendments with *Medicago* plant materials on the non-parasitic, beneficial components of soil nematode community. Therefore, a study was carried out to assess the short-time effects of soil amendments with biomasses from the same six different *Medicago* species as previously tested on the root-knot nematode *M. incognita*, i.e., *M. sativa*, *M. heyniana*, *M. hybrida*, *M. lupulina*, *M. murex*, and *M. truncatula* [17], on the overall nematofauna of natural tomato-planted soil. 

## 2. Results

### 2.1. Nematode Abundance and Biomass

Nematode abundance was differently affected by the six *Medicago* plant biomasses´ incorporation (Figure 1A). Compared to the non-amended control, a significantly lower abundance was observed in soil amended with *M. heyniana*, *M. hybrida*, and *M. lupulina*. In contrast, the incorporation of the biomasses from *M. murex*, *M. sativa*, and *M. truncatula* resulted in no significant impact on overall nematode abundance. No significant variation of total nematode individuals occurred among the different amendment rates, except for the highest rate of *M. lupulina*. Adversely to abundance, all amendments with the *Medicago* plant materials resulted in a significant decrease of the total nematode biomass, irrespective of the applied incorporation rate (Figure 1B).

### 2.2. Nematode Trophic Groups

Nematofauna of non-amended soil was prevalently constituted by bacterivores (41.4%), plant parasitic nematodes (33.7%), and fungivores (14.6%), with a low presence of root-fungal feeders (5.4%), omnivores (3.9%), and predators (Figure 2). Relative abundance of bacterivores was significantly increased by the incorporation of biomasses from all the *Medicago* species except *M. truncatula*. Adversely, only *M. truncatula* amendments resulted in a significant increase of herbivores, i.e., plant parasites, as significantly reduced by all the other *Medicago* biomasses. Relative abundance of fungivores significantly raised in soil amended with *M. heyniana*, *M. lupulina*, and *Megicago sativa* plant materials, but was not statistically affected by the other three *Medicago* species. Only *M. heyniana* and *M. lupulina* amendments negatively affected the abundance of omnivores, whereas all six species biomasses significantly reduced the incidence of root fungal feeders and poorly influenced the presence of predators. All the recorded effects were not significantly related to the amendment rates.

### 2.3. Nematode Genera

A total of 31 nematode genera were identified in non-treated soil, whereas this number ranged from a minimum of 22 (treatment with 10 g kg^−1^ soil of *M. lupulina* biomass) to 30 (*M. sativa* at 10 g kg^−1^ soil) in the amended soil (Appendix A). Bacterivore genera were the most numerous (15), followed by omnivores (5), fungivores, as well as plant parasites and predators (3), whereas only two genera of root-fungal feeders were identified. Within bacterivores, *Acrobeloides*, *Cephalobus*, *Plectus*, *Mesorhabditis*, and *Rhabditis* were the most abundant genera, whereas *Aphelenchoides* and *Aphelenchus* prevailed among the fungivores.

All the tested amendments increased the abundance of only two nematode genera, the bacterivore *Rhabditis* and the fungivore *Aphelenchus*. Among plant parasites, the population of genus *Meloidogyne* was significantly reduced, whereas that of *Geocenamus* strongly increased in soil amended with *M. lupulina*, *M. murex*, and *M. truncatula* biomasses, as giving account of the highest nematode abundance recorded for the 10 g kg^−1^ soil treatment with the *M. truncatula* biomass. 

### 2.4. Soil Food Web Indices

Compared to control, the Maturity Index (MI) and Sum Maturity Index (∑MI) were significantly higher in soil treated with *M. heyniana*, *M. hybrida*, *M. lupulina*, and *M. sativa* plant material (Table 1). 

Adversely, the values of Plant Parasitic Index (PPI) and MI2-5 were not significantly affected by any soil amendment with *Medicago* plant biomass. The Channel index (CI) significantly decreased in pots treated with *M. heyniana*, *M. hybrida*, *M. sativa*, and *M. truncatula*, while the Enrichment index (EI) resulted in an increase by the amendments with plant materials from all the *Medicago* species except for *M. murex*. 

Significant differences among the non-treated control and amended soil were also found for the values of nematode metabolic footprints (Table 2). In particular, soil amendments with *Medicago* plant materials always resulted in a significant dose-related decrease of composite and fungivore footprints and, adversely, in significantly higher values of enrichment and bacterivore footprints.

## 3. Discussion

Nematodes play a key role within the soil food web due to their strict relationship with soil microorganisms and their functions [24,31,39]. In consideration of this key position, the analysis of soil nematofauna has been largely acknowledged as an effective tool for the assessment of soil health and quality status following chemical, physical, and agricultural perturbations, also including soil organic amendments [33,35,40]. 

In this study, soil incorporation with the biomasses of the six *Medicago* plants caused an overall decrease of nematofauna abundance and biomass. Contrastingly, most literature studies generally reported an increase of total nematode abundance following soil amendments with various organic materials [12,40,41,42,43,44]. However, the observed decrease of nematofauna abundance and biomass was mainly related to the strong suppression of plant parasites, while beneficial nematofauna components such as bacterivores, fungivores, and omnivores were almost generally increased. This is the first report of the effects of soil amendments with *Medicago* plant materials on the whole soil nematofauna, as the previous studies were limited to the assessment of the impact of various *Medicago* plant biomasses on selected plant-parasitic nematode species [14,15,16,17].

The increase of bacterivore abundance observed in our study is in good agreement with literature data, which reported a larger presence of bacterivore and fungivore nematofauna after soil amendments with biomasses from other Leguminosae plants, such as vetch or velvet bean [36,37]. A bacterial decomposition pathway was also observed in soil incorporated with green biomasses of Brassicaceae plants as white mustard and radish [12], whereas the prevalence of a fungal-based food web was observed in soil amended with rapeseed (*Brassica napus* L.) and rye [38]. The large feeding substrate provided to bacterial and fungal populations by the incorporated *Medicago* biomasses can reasonably explain the shift of soil nematofauna towards the bacterivore and fungivore trophic groups, as also stated by previous literature studies [29,45,46,47,48]. An increased populations of bacterivorous and fungivorous nematodes was assessed also after soil amendments with other Leguminosae plants, such as vetch or velvetbean [36,37]. Genera *Rhabditis*, *Mesorhabditis*, *Acrobeloides*, *Plectus*, and *Cephalobus* were the most abundant among bacterivores, whereas the species of *Aphelenchus* and, at a lesser extent, *Aphelenchoides* prevailed among the fungivores. A prevalence of Rhabditidae and Cephalobidae bacterivores was also observed in the studies of Bulluck et al. [36] and Forge et al. [49], as well as Porazinska et al. [50] describing a dramatic increase of Rhabditidae bacterivores following to compost amendments in orchard soils. Moreover, McSorley and Frederick [37] reported a prevalence of genera *Aphelenchoides* and *Aphelenchus* within the fungivore nematofauna also following soil amendments with velvet bean biomass. Analogously, our findings of an increased abundance of omnivores, as well as of limited or nil effects on predators, are also corroborated by the the results from previous studies in fields amended with various organic materials [12,47,48]. As indicated for bacterivores, shifts of omnivore and predator populations should be related to the alteration of their food sources more than to a direct toxic effect of *Medicago* plant materials [40,43]. Finally, the strong suppressing effect showed by the biomasses from *M. heyniana*, *M. hybrida*, *M. murex*, and *M. sativa* on plant-parasitic nematofauna fits well the results from our previous comparative study on *M. incognita* on tomato [17], though in the absence of the direct relationship of suppressive effects with the amendment rates. However, the generation time of a majority of free-living nematode genera is still unknown, as it is very difficult or almost impossible to determine, and all the characteristics of feeding groups or c-p gropus are assumptions based on ecological and/or nematological studies [21,24,28,31,32]. Consequently, we do not know if the lifecycle of the found nematode genera correspond to our experimental time, i.e., they all were able to reproduce in that time frame or if that could have affected the final results. Therefore, our considerations are necessarily limited to the short-time impact of the tested amendments on the detected nematode genera, while further time-extended studies should be needed to assess the long period population dynamics. Nematode community indices were developed as synthetic indicators of the status of soil food web as well as of the soil environment [24]. Values of Maturity Indices (MI, MI2-5 and ∑MI) are generally reduced by any shifts of soil nematofauna towards more adaptable nematode species, i.e., species with a high reproductive activity, short life cycle, and high tolerance to changes of soil conditions (e.g., Rhabditidae, Panagrolaimidae), such as those occurring after organic amendments. In good agreement with data from our current study, lower values of maturity indices in soil amended with various organic matrices than in non-treated soil were also documented by previous literature studies [36,42,46].

The Enrichment Index (EI) is based on the expected responsiveness of the opportunistic guilds (bacterivorous nematodes with c-p value equal to one) to organic resource enrichment. Therefore, EI describes whether the soil food web is nutrient enriched (high EI) or depleted (low EI) [24]. On the other hand, values of Channel Index (CI) indicate the predominant decomposition channel in the soil food web. A high CI (over 50%) indicates a higher proportion of fungal decomposition and reflect the high relative abundance of c-p fungal feeders (Aphelenchoididae and Aphelenchidae) and the corresponding low abundance of c-p1 bacterial feeders (Rhabditidae and Panagrolaimidae), whereas a low CI (under 50%) suggests a bacterial decomposition channel. Literature studies generally described an EI increase and a CI decrease in soils amended with organic materials compared to non-amended control, due to a predominant bacterial decomposition channel [12,51,52,53]. Therefore, the low values of CI and high values of EI, as well as the significant decrease of the values of all maturity indices, recorded in soil amended with most of *Medicago* plant biomasses indicate that also these materials can nutritionally enrich the soil environment and thus support the bacterial decomposition channel, despite the increased number of fungivores in amended soils.

A calculation of the metabolic footprint of different nematode trophic groups enables a functional quantification of biomass, metabolic activity, and magnitude of carbon and energy flow occurring in the soil food web through their respective channels [32,54]. In this study, values of metabolic footprints indicate that most C flow in soil amended with *Medicago* plants biomasses occurred through the bacterial channel. However, there is also a relevant C flow through the fungal channel in soil amended with *M. lupulina* and *M. sativa* biomasses, as indicated by the significantly higher fungivore footprint and density.

## 4. Materials and Methods 

### 4.1. Experimental Design 

Natural sandy soil (64.4% sand,18.7% silt, 16.9% clay, 0.8% OM, pH 7.5) was collected from a field located at Castellaneta (province of Taranto, Apulia region). The soil was thoroughly mixed and treated with 10, 20, or 40 g kg^−1^ soil rates of the same batches of dry green biomasses of *M. heyniana*, *M. hybrida*, *M. lupulina*, *M. murex*, *M. sativa*, and *M. truncatula* used in the previous study by D’Addabbo et al. [17]. The amended soil was then poured into 1.5 L clay pots, providing five replicates for each treatment and including non-treated soil as a control. As in D’Addabbo et al. [17], a one-month-old tomato seedling (cv Tomito) was transplanted in each pot two weeks after the soil incorporation with the *Medicago* plant biomasses. Plants were maintained in a glasshouse (25 ± 2 °C constant temperature) for a two month period. The experiments were repeated twice.

### 4.2. Nematode Extraction and Identification

A composite 100 mL soil sample was collected from each pot and soaked for 30 min in 1 L of tap water. Samples were then processed by a combination of Cobb’s sieving and decanting method [55] and a modified Baermann technique [56]. The extracted nematodes were at first examined under a stereomicroscope and then fixed with a hot 99:1 solution of 4% formaldehyde: pure glycerol. Fixed nematodes were microscopically (Nikon Eclipse 90i light microscope) identified to a genera level according to original species descriptions and taxonomic keys [57,58,59,60]. Nematode abundance in each sample was expressed as the number of individuals per 100 g of dry soil after gravimetrically measuring soil moisture content by oven drying (105 °C for 24 h) 100 g of soil to constant weight.

### 4.3. Nematode Identification and Classification

Nematode genera were partitioned to 6 trophic groups, i.e., bacterivores (B), fungivores (F), root-fungal feeders (facultative plant parasites) (RFF), obligatory plant parasites (PP), predators (P), and omnivores (O), according to Yeates et al. [21] and Wasilewska [61] and following adjustments and supplementations of Sieriebriennikov et al. [62]. Genera were also assigned to the colonizer-persister (c-p) 1–5 scale based on their r and k characteristics, according to Bongers and Bongers [28] and Bongers [31]. In particular, C-p1 taxa consist of “r-strategists”, with short generation times, small eggs, and high fecundity, whereas c-p5 taxa consist of “k-strategists”, with long generation times, large bodies, low fecundity, and high sensitivity to disturbance [24]. The total number of genera, total nematode abundance, relative abundance of nematodes per trophic group, and nematode biomass were also determined. 

### 4.4. Ecological and Functional Indices

Changes in the nematode communities were also evaluated by calculating different ecological and functional indices and metabolic footprints. Ecological indices as maturity index MI [31], sum maturity index ∑MI [63], maturity index MI2-5 [63] for non-parasitic nematodes and plant parasitic index PPI [31], functional indices enrichment index EI and channel index CI (24), nematode biomass by Andrassy’s formula [64], and metabolic footprints were calculated using the NINJA online software [62]. 

### 4.5. Statistical Analysis 

Data from the two runs of the experiment were pooled as no significant experiment x treatment interactions were found [65]. The pooled data were arcsin-transformed as to homogenize error variances and then subjected to one-way analysis of variance, comparing means by the Least Significant Difference Test at *p* ≤ 0.05 [65]. All statistical analyses were performed by the PlotIT 3.2 (Scientific Programming Enterprises, Haslett, MI, USA) software. 

## 5. Conclusions

This study indicated that soil amendments with biomasses from *Medicago* plants could not only suppress the plant-parasitic nematodes but also increase the beneficial components of the soil nematode community, such as the bacterivore and fungivore species. *Medicago* materials seem to be particularly suitable to nematicidal soil amendments, as operating a biological soil disinfestation from plant parasites due to their high content of bioactive saponins and the release of biocidal ammoniacal nitrogen [16], while acting at the same time as a food source for beneficial trophic groups. More generally, the association of reduced plant-parasitic nematode infestations and related soilborne fungal diseases to an improved soil quality could result in better crop growth and crop yield, as also confirmed by previous experiments in the field [15]. According to these overall effects, soil amendments with *Medicago* plant materials or their formulated derivatives could be suggested as an additional tool for a sustainable control phytonematode.

## Figures and Tables

**Figure 1 plants-10-00145-f001:**
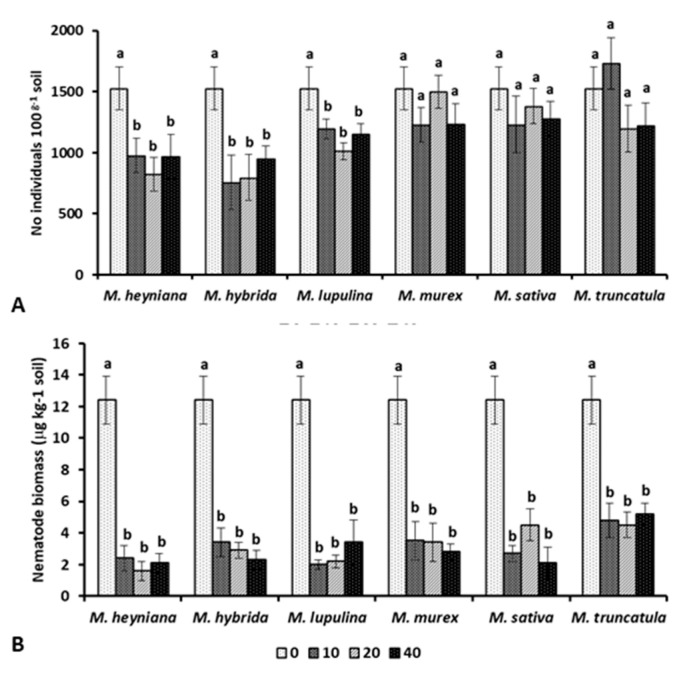
Total nematode abundance (**A**) and biomass (**B**) in soil amended with 10, 20, or 40 g kg^−1^ soil rates of dry plant biomass of six *Medicago* plant species.

**Figure 2 plants-10-00145-f002:**
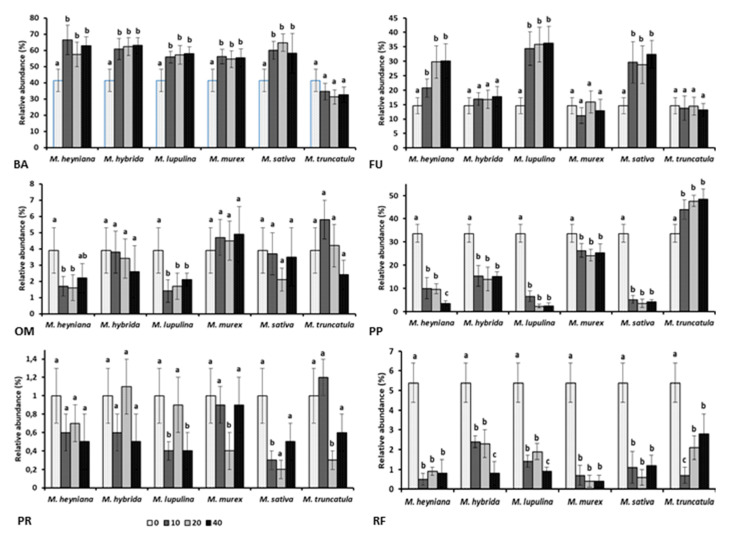
Relative abundance of the different nematode trophic groups in soil amended with 10, 20, or 40 g kg^−1^ soil rates of dry plant biomass from six different *Medicago* species. BA = bacterivores; FU = fungivores; OM = omnivores; PP = plant parasites; PR = predators; and RF = root fungal feeders.

**Table 1 plants-10-00145-t001:** Soil food web descriptors in soil amended with dry biomasses from six *Medicago* species.

Rate (g kg^−1^ soil)	MI		PPI		MI 2–5		∑MI		CI		EI	
*M. heyniana*												
0	2.09 ± 0.05	a	2.96 ± 0.05	a	2.21 ± 0.07	a	2.48 ± 0.09	a	48.9 ± 15.6	a	43.9 ± 8.46	a
10	1.47 ± 0.11	b	3.00 ± 0.00	a	2.17 ± 0.14	a	1.62 ± 0.11	b	9.6 ±4.7	b	86.8 ± 5.6	b
20	1.66 ± 0.5	b	2.96 ± 0.02	a	2.07 ± 0.03	a	1.82 ± 0.12	b	18.5 ± 4.5	b	75.4 ± 4.1	b
40	1.59 ± 0.12	b	3.00 ± 0.00	a	2.12 ± 0.04	a	1.64 ± 0.18	b	14.9 ± 5.0	b	81.2 ± 5.1	b
*M. hybrida*												
0	2.09 ± 0.05	a	2.96 ± 0.05	a	2.21 ± 0.07	a	2.48 ± 0.09	a	48.9 ± 15.6	a	43.9 ± 8.46	a
10	1.53 ± 0.28	b	2.98 ± 0.02	a	2.42 ± 0.09	b	1.76 ± 0.23	b	7.6 ± 3.0	b	90.2 ± 3.9	b
20	1.71 ± 0.22	a	2.98 ± 0.02	a	2.29 ± 0.03	a	1.89 ± 0.22	b	17.4 ± 8.5	b	80.3 ± 8.6	b
40	1.69 ± 0.15	b	2.99 ± 0.02	a	2.11 ± 0.07	a	1.77 ± 0.17	b	14.5 ± 5.2	b	73.2 ± 10.4	b
*M. lupulina*												
0	2.09 ± 0.05	a	2.96 ± 0.05	a	2.21 ± 0.07	a	2.48 ± 0.09	a	48.9 ± 15.6	a	43.9 ± 8.46	a
10	1.84 ± 0.01	a	3.00 ± 0.00	a	2.05 ± 0.03	a	1.91 ± 0.05	b	31.9 ± 4.3	a	60.6 ± 2.1	b
20	1.79 ± 0.05	b	2.88 ± 0.04	a	2.08 ± 0.04	a	1.82 ± 0.05	b	27.4 ± 7.9	b	66.8 ± 5.3	b
40	1.79 ± 0.16	b	3.00 ± 0.00	a	2.12 ± 0.07	a	1.82 ± 0.15	b	31.1 ± 13.3	a	67.8 ± 12.2	b
*M. murex*												
0	2.09 ± 0.05	a	2.96 ± 0.05	a	2.21 ± 0.07	a	2.48 ± 0.09	a	48.9 ± 15.6	a	43.9 ± 8.46	a
10	2.08 ± 0.05	a	3.00 ± 0.00	a	2.27 ± 0.09	a	2.36 ± 0.12	a	32.8 ± 13.7	a	39.9 ± 9.7	a
20	2.03 ± 0.07	a	2.95 ± 0.07	a	2.12 ± 0.07	a	2.24 ± 0.07	a	48.8 ± 9.2	a	41.4 ± 8.2	a
40	2.09 ± 0.06	a	2.96 ± 0.02	a	2.17 ± 0.07	a	2.32 ± 0.04	a	40.8 ± 14.1	a	34.9 ± 9.8	a
*M. sativa*												
0	2.09 ± 0.05	a	2.96 ± 0.05	a	2.21 ± 0.07	a	2.48 ± 0.09	a	48.9 ± 15.6	a	43.9 ± 8.46	a
10	1.96 ± 0.09	a	2.98 ± 0.02	a	2.13 ± 0.04	a	2.01 ± 0.15	b	36.5 ± 9.2	a	52.5 ± 9.7	a
20	1.69 ± 0.06	b	2.96 ± 0.02	a	2.09 ± 0.04	a	1.73 ± 0.06	b	17.2 ± 4.2	b	74.4 ± 4.7	b
40	1.86 ± 0.14	a	2.98 ± 0.04	a	2.14 ± 0.09	a	1.91 ± 0.14	b	27.9 ± 12.6	b	64.9 ± 6.6	b
*M. truncatula*												
0	2.09 ± 0.05	a	2.96 ± 0.05	a	2.21 ± 0.07	a	2.48 ± 0.09	a	48.9 ± 15.6	a	43.9 ± 8.46	a
10	1.94 ± 0.39	a	3.00 ± 0.00	a	2.36 ± 0.30	a	2.41 ± 0.21	a	19.2 ± 11.9	b	72.5 ± 8.3	b
20	1.91 ± 0.15	a	2.99 ± 0.02	a	2.29 ± 0.12	a	2.42 ± 0.21	a	22.0 ± 5.8	b	70.5 ± 6.1	b
40	1.71 ± 0.13	b	3.00 ± 0.00	a	2.25 ± 0.09	a	2.33 ± 0.11	a	15.5 ± 5.1	b	78.8 ± 8.4	b

Each value is a mean of five replications. Data marked with the same letter in each column are not statistically different to untreated control according Least Significant Difference Test (*p* = 0.05). MI = Maturity index; PPI = Plant parasitic index; MI2-5 = Maturity index for colonizers-persisters group 2–5; ∑MI = Sum Maturity index; CI = Channel index; EI—Enrichment index.

**Table 2 plants-10-00145-t002:** Nematode metabolic footprints (means ± SE) after soil treatment with dry biomass from six *Medicago* species.

Rate (g kg^−1^ soil)	Cfoot		Efoot		Hfoot		Ffoot		Bfoot		Pfoot		Ofoot	
*M. heyniana*														
0	2027.6 ± 158.9	a	78.5 ± 23.3	a	1807.2 ± 92.4	a	18.8 ± 5.7	a	124.2 ± 37.9	a	11.6 ± 3.6	a	65.9 ± 12.7	a
10	573.5 ± 58.8	b	303.8 ± 52.4	b	237.5 ± 31.8	b	20.2± 2.4	a	302.9 ± 69.7	b	3.6 ± 2.1	a	9.5 ± 4.6	b
20	354.2 ± 18.8	b	177.3 ± 28.8	b	127.5 ± 53.8	b	24.9 ± 8.1	a	175.9 ± 55.8	a	1.8 ± 0.6	a	23.9 ± 14.2	b
40	492.8 ± 44.4	b	317.5 ± 44.7	b	109.0 ± 13.7	b	29.9 ± 6.4	b	307.7 ± 94.4	b	3.2 ± 2.0	a	43.6 ± 17.8	ab
*M. hybrida*														
0	2027.6 ± 158.9	a	78.5 ± 23.3	a	1807.2 ± 92.4	a	18.8 ± 5.7	a	124.2 ± 37.9	a	11.6 ± 3.5	a	65.9 ± 1 2.7	a
10	648.5 ± 88.7	b	193.8 ± 61.1	b	391.1 ± 45.6	b	11.2 ± 2.1	b	189.9 ± 38.88	b	3.5 ± 1.8	a	52.8 ± 23.1	a
20	558.5 ± 55.9	b	186.8 ± 56.8	b	297.5 ± 77.2	b	20.2 ± 3.9	a	183.0 ± 67.36	b	9.4 ± 2.6	b	48.5 ± 15.2	a
40	488.8 ± 35.4	b	207.6 ± 33.8	b	232.9 ± 55.7	b	20.4 ± 4.0	a	226.8 ± 77.1	b	3.0 ± 1.2	a	14.3 ± 3.2	b
*M. lupulina*														
0	2027.6 ± 158.9	a	78.5 ± 23.3	a	1807.2 ± 92.4	a	18.8 ± 5.7	a	124.2 ± 37.9	a	11.6 ± 3.5	a	65.9 ± 12.7	a
10	506.3 ± 69.9	b	350.2 ± 88.2	b	80.8 ± 12.8	b	36.5 ± 11.3	b	368.9 ± 83.2	b	5.1 ± 3.0	b	15.0 ± 5.9	b
20	514.5 ± 60.3	b	392.8 ± 69.4	b	60.2 ± 10.7	b	32.6 ± 10.8	b	403.7 ± 61.2	b	6.5 ± 1.5	b	11.8 ± 6.8	b
40	879.7 ± 55.6	b	745.3 ± 101.2	b	62.3 ± 30.8	b	50.1 ± 8.3	b	732.4 ± 115.2	b	6.9 ± 2.5	b	28.3 ± 12.4	b
*M. murex*														
0	2027.6 ± 158.9	a	78.5 ± 23.3	a	1807.2 ± 92.4	a	18.8 ± 5.7	a	124.2 ± 37.9	a	11.6 ± 3.6	a	65.9 ± 12.7	a
10	698.8 ± 64.5	b	115.8 ± 23.9	b	418.7 ± 115.9	b	13.3 ± 5.4	a	197.8 ± 52.3	b	10.2 ± 4.2	b	58.6 ± 10.2	a
20	681.4 ± 47.7	b	154.5 ± 30.6	b	330.3 ± 86.1	b	34.8 ± 12.8	b	215.0 ± 36.9	b	3.5 ± 1.2	a	97.9 ± 18.2	b
40	573.8 ± 88.2	b	122.1 ± 54.7	b	240.3 ± 52.4	b	16.4 ± 3.9	a	204.5 ± 54.5	b	11.5 ± 3.7	b	101.1 ± 31.3	b
*M. sativa*														
0	2027.6 ± 158.9	a	78.5 ± 23.3	a	1807.2 ± 92.4	a	18.8 ± 5.7	a	124.2 ± 37.9	a	11.6 ±3.6	a	65.9 ± 12.7	a
10	635.8 ± 66.2	b	374.7 ± 55.7	b	107.5 ± 27.2	b	35.8 ± 6.7	b	422.1 ± 69.4	b	2.3 ± 0.6	b	67.2 ± 20.2	a
20	1152.8 ± 114.8	b	972.4 ± 83.7	b	93.8 ± 15.5	b	35.0 ± 8.2	b	998.0 ± 155.5	b	1.3 ± 0.2	b	24.7 ± 6.7	b
40	518.7 ± 99.1	b	348.0 ± 69.3	b	73.9 ± 29.6	b	23.4 ± 6.9	a	377.9 ± 57.2	b	2.8 ± 1.0	b	41.0 ± 11.1	b
*M. truncatula*														
0	2027.6 ± 158.9	a	78.5 ± 23.3	a	1807.2 ± 92.4	a	18.8 ± 5.7	a	124.2 ± 37.9	a	11.6 ± 3.6	a	65.9 ± 12.7	a
10	1034.5 ± 42.6	b	394.9 ± 51.1	b	496.2 ± 60.2	b	23.9 ± 3.7	a	417.9 ±36.8	b	11.2 ± 3.1	a	85.5 ± 21.1	a
20	910.1 ± 63.8	b	304.3 ± 65.2	b	541.8 ± 98.3	b	16.5 ± 5.6	a	318.3 ± 48.3	b	3.2 ± 1.2	b	31.6 ± 8.3	b
40	1040.8 ± 86.4	b	389.1 ± 12.7	b	564.0 ± 61.3	b	14.9 ± 5.5	a	394.4 ± 63.7	b	5.5 ± 1.7	b	62.1 ± 5.9	a

Each value is a mean of five replications. Data marked with the same letter in each column are not statistically different to untreated control according Least Significant Difference Test (*p* = 0.05). Cfoot = Composite footprint; Efoot = Enrichment footprint; Hfoot = Herbivore footprint; Ffoot = Fungivore footprint; Bfoot = Bacterivore footprint; Pfoot = Predator footprint; Ofoot = Omnivore footprint.

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
