# Peer review of "Short-Time Impact of Soil Amendments with Medicago Plant Materials on Soil Nematofauna"

_plants, 2021, doi:10.3390/plants10010145_

Round 1

Reviewer 1 Report

This manuscript describes the impact of soil amendments from six different species of Medicago on soil nematode communities, in a pot assay with tomato plants, in a glasshouse. Nematodes are important bioindicators of soil quality and health as they respond rapidly to enrichment and disturbance. Due to the negative economic impact of plant-parasitic nematodes (PPN) and the increasing need to develop more sustainable cultural practices (i.e. those that promote plant health by minimizing the negative impact of PPN whilst sustaining non-target nematodes diversity and abundance), studies that focus on the effects of sustainable PPN management tools, such as organic soil amendments, on soil nematode diversity and abundance are of uttermost importance.

The article is well structured, in a clear and eloquent approach that allows the reader to understand the importance of the study, the methodologies carried out and, the results and conclusions are consistent and adequate.

Minor details:

Lines 64-67: please rewrite. There is a repetition of plant parasitism as a feeding behaviour; “in addition to” and then it is cited again at the end of the sentence. Suggestion: begin sentence with “Components of soil nematofauna .....”

Line 101: Meloidogyne and Medicago sativa. The genus names should be written in full in this sentence to avoid risk of confusing both genera.

Line 237: Meloidogyne incognita (see comment above)

Figure 1. indicate soil rates of dry plant biomass in the legend

Table I – suggestion: include as supplemental material; also place names of trophic groups with Genera; align the first column

Author Response

We would like to thank the independent reviewer for the evaluation and appreciation of our work

Please see the attachment to see our changes base of reviewer proposal

Table 1 was accepted and is now as supplementary table, therefore was transfered at the end of tamplate. 

The table S1 is attached also as separate file under second reviewer 

Reviewer 2 Report

This is a very interesting paper dealing with the influence of soil amendments with Medicago species on the overall nematofauna. The manuscript presents new and significant information these amendments on several nematode trophic groups, deserving publication in Plants. The paper is very well-organized and has been carried out following all the international standards and deserves publication in the journal Plants. I include below two minor editorial mistakes.

  1. Please replace plant parasitic by plant-parasitic throughout all the manuscript
  2. L106-116, I suggest to include here the percentage of decreasing in nematode abundance for each Medicago species, heyniana, M. hybrida, M. lupulina.
  3. L114-115, replace materals by materials
  4. For a better understanding to general readers, in footnotes of Tables 2 and 3, please add the reference/s from which these indexes were described.
  5. L285 replace study of.... by study by D'Addabbo
  6. L290 replace was by were
  7. L311, please indicate the formula and reference used for calculating nematode biomass
  8. L331 replace thanks by due
  9. L338 replace sustainable by sustainable control
  10. Please remove () from references numbers 41, 50, 53, 54

Author Response

We would like to thank the independent reviewer for the evaluation and appreciation of our work

Please see the attachment to see our changes base of reviewer proposal

Authors,

Reviewer 3 Report

This is a well prepared manuscript presenting new and valuable information on "Short-time impact of soil amendments with Medicago plant materials on soil nematofauna". This study is very well carried out with promising results worthy of publication. All amendments resulted in a significant decrease of the total nematofauna biomass, whereas total abundance was significantly reduced only by M. heyniana, M. hybrida and M. lupulina biomasses. Almost all the Medicago amendments significantly reduced the relative abundance of plant parasites and root fungal feeders. All amendments significantly increased abundance of bacterivores, whereas fungivores significantly increased only in soil amended with M. heyniana, M. lupulina and M. sativa plant materials. Interesting to note was that Mesorhabditis and Rhabditis were the most abundant genera of bacterivores, whereas Aphelenchoides and Aphelenchus prevailed among the fungivores. Predators were poorly influenced by all the tested Medicago biomasses, whereas abundance of omnivores was negatively by M. heyniana and M. lupulina. Values of Maturity Index and Sum Maturity Index were reduced by treatments with M. heyniana, M. hybrida, M. lupulina and M. sativa plant materials, whereas most of the tested amendments decreased values of Channel Index while increasing those of Enrichment Index. Enrichment and bacterivore footprints raised following to soil addition with Medicago biomasses, whereas composite and fungivore footprints were significantly reduced. Certainly, their overall positive effects on soil nematofauna, amendments with Medicago plant materials or their formulated derivatives could represent an additional tool for a sustainable management of phytoparasitic nematodes. Excellent studies including data given in the form of Figures and Tables worth publishing!

Author Response

We would like to thank the independent reviewer for the appreciation of our work

Authors,